# $\mathcal{N}$ Attack: A Strong and Universal Gaussian Black-Box Adversarial Attack

## Abstract

Recent works find that DNNs are vulnerable to adversarial examples, whose changes from the benign ones are imperceptible and yet lead DNNs to make wrong predictions. One can find various adversarial examples for the same input to a DNN using different attack methods. In other words, *there is a population of adversarial examples, instead of only one, for any input to a DNN*. By explicitly modeling this adversarial population with a Gaussian distribution, we propose a new black-box attack called $\mathcal{N}$ ATTACK. The adversarial attack is hence formalized as an optimization problem, which searches the mean of the Gaussian under the guidance of increasing the target DNN's prediction error. $\mathcal{N}$ ATTACK achieves 100% attack success rate on six out of eleven recently published defense methods (and greater than 90% for four), all using the same algorithm. Such results are on par with or better than powerful state-of-the-art white-box attacks. While the white-box attacks are often model-specific or defense-specific, the proposed black-box $\mathcal{N}$ ATTACK is universally applicable to different defenses.

## 1 Introduction

Deep neural networks (DNNs) have triumphed over many perception and control tasks for which there exist big (labeled) data to train the networks. Phenomenal examples include that a DNN-powered agent beats world champions on playing the game of Go (Silver et al., 2016), large-scale image classification (Russakovsky et al., 2015; Krizhevsky et al., 2012; He et al., 2016), acoustic speech recognition (Hinton et al., 2012), *etc.* As a result, DNNs are more and more widely used in real products (*e.g.*, self-driving, face recognition, Amazon Go, and Web content understanding).

This paper is concerned with the security aspect of DNNs. We aim to provide a *strong* adversarial attack method which can *universally* attack a variety of DNNs and defense techniques. Progress on this will significantly facilitate the research on robust DNNs to be deployed in uncertain and even adversarial environments.

The seminal work by Szegedy et al. (2013) finds that DNNs are vulnerable to *adversarial examples*, whose changes from the benign ones are imperceptible and yet can manipulate a DNN and lead to wrong predictions. A rich line of works furthering their finding reveals more worrisome results. Notably, adversarial examples are transferable, meaning that one can design adversarial examples for one DNN and then use them to fail others (Papernot et al., 2016a; Szegedy et al., 2013; Tramèr et al., 2017b). Moreover, Moosavi-Dezfooli et al. (2017) show that a single perturbation pattern may convert a large number of test images into adversarial ones. Finally, recent works (Carlini & Wagner, 2017; Athalye et al., 2018) have defeated several defenses against earlier adversarial attack approaches.

It remains unclear what causes DNNs severely sensitive to adversarial examples. Goodfellow et al. (2014b) conjecture that DNNs behave linearly in the high dimensional input space, amplifying small perturbations when their signs clone the signs of the DNNs' intrinsic linear weights. Fawzi et al. (2018) experimentally study the topology and geometry of adversarial examples. Ma et al. (2018) characterize the subspace of adversarial examples. Nonetheless, defense methods (Papernot et al., 2015; Tramèr et al., 2017a; Rozsa et al., 2016; Madry et al., 2018) motivated by them are broken in a relatively short amount of time (He et al., 2017; Athalye et al., 2018; Xu et al., 2017; Sharma & Chen, 2017), implying that either better defense techniques are yet to be developed or alternative factors are still not identified that contribute to the the sensitivity of DNNs.

With no doubt, toward robust DNNs there is a pressing need to gain a good understanding about the mechanism how adversarial examples fool DNNs. Apart from that, it is also vital to empirically devise defense methods and test them against adversarial attacks on benchmark datasets. The latter, however, is impeded by the fact that existing adversarial attacks are either model- and defense-specific (*e.g.*, white-box attack) or relatively weak (*e.g.*, black-box attack).

White-box attack assumes that it has full knowledge of a DNN (network architecture, weights, input and output spaces, *etc.*) and produces adversarial examples by propagating gradients of certain loss back to inputs. When some defense methods "obfuscate" gradients, Athalye et al. (2018) find ways to approximate the gradients which still give rise to extremely high attack success rates. Despite being powerful, white-box attack methods are often model-specific. Consequently, when a new defense method is proposed, one may not be able to test its performance by using existing white-box attacks. Indeed, Buckman et al. (2018) have to derive a special attack alongside their thermometer-encoding based defense in order to verify the latter's effectiveness.

In contrast, black-box attacks are universally applicable as they do not rely on the networks' architectures or weights at all. Such methods compose adversarial examples by tracking how the change to the input affects the output of a DNN (Papernot et al., 2017; Chen et al., 2017). However, existing black-box attacks have been mainly tested on vanilla DNNs, leaving it unclear how strong they are at attacking the defended DNNs. Our experiments show that ZOO (Chen et al., 2017), a prevalent black-box attack method, fails under many defense techniques. The decision-based method (Brendel et al., 2017) gives rise to zero success rate at attacking (Guo et al., 2018). Resistance to the black-box attacks does not guarantee a success in defending DNNs against white-box attacks.

**Our approach.** In this paper, we propose a Gaussian black-box adversarial attack ($\mathcal{N}$ATTACK) whose performance is as *strong* as the existing white-box attacks, plus being *universal*. The main idea draws upon the fact that one can find various adversarial examples for the same input to a DNN by using different attack methods. In other words, **there is a population of adversarial examples, instead of only one, for any input to a DNN.** We model this adversarial population by a Gaussian distribution. The adversarial attack is then formalized into an optimization problem which searches for the mean of the Gaussian under the guidance of increasing the target DNN's prediction error. We solve the problem by an evolution strategy (ES) (Wierstra et al., 2008; Salimans et al., 2017).

The resulting algorithm is coincidentally close to the one developed by Ilyas et al. (2018). Unlike ours, Ilyas et al. (2018) use ES in order to estimate the gradients in the white-box PGD attack (Madry et al., 2018) — in essence, it is in the same vein as the gradient estimation work (Athalye et al., 2018). Consequently, their approach heavily depends on the quality of the estimated gradients. When the gradients are "obfuscated", ES cannot well approximate them, giving rise to low attack success rates (cf. Section 3.1.3). To alleviate the dependence on the gradients, we do not employ any white-box attack methods at all and, instead, model the whole population of adversarial examples for every single image by a Gaussian distribution. The Gaussian mean is more important than the gradients in our approach. While ES is employed to search for the Gaussian mean in this work, as opposed to approximating the gradients of PGD, the other derivative-free methods (Rios & Sahinidis, 2013) are also applicable and are left for future work.

The initialization to the population mean plays a key role in our approach. Given a good initialization, the evolution strategy can quickly find adversarial examples. We train a regression neural network which takes as input the input to the target DNN and outputs an adversarial perturbation. Whereas the regressed example can rarely defeat the target DNN, it serves as a good starting point for the optimization algorithm and hence speeds up the $\mathcal{N}$ATTACK.

We achieve 100% attack success rate on six out of eleven recently published defense methods (and greater than 90% on four), all using the same algorithm. On the contrary, white-box attacks employ distinct defense-specific gradient approximations (Athalye et al., 2018) or particularly tailored attacks (Buckman et al., 2018). We expect the proposed $\mathcal{N}$ATTACK, which is both strong and universal, can serve as a convenient baseline, along with the ROC curve described below, to facilitate the future research toward robust DNNs.

**A new curve for evaluation.** The evolution strategy we used in the paper resembles the policy gradient theorem (Salimans et al., 2017) — supplied with sufficient computation resources, it is guaranteed to converge to a local optimum. In expectation, the attack strength of the adversarial

population grows at every iteration of the optimization algorithm. This property is very appealing, as it gives rise to another perspective to analyze the robustness of a defense method. To this end, we propose a new curve featuring the attack success rate versus number of evolution iterations — different strengths of the $\mathcal{N}$ATTACK. The curve complements the attack success rate by revealing the dynamics of a defense method. One can conveniently read from the curve, to what attack strength or number of DNN queries the attack makes, a defense method is able to resist if one sets a robustness threshold for the attack success rate.

## 2    APPROACH

Consider a DNN classifier $C(x) = \arg\max_i F(x)_i$, where $x \in [0, 1]^D$ is an input to the neural network $F(\cdot)$. Without loss of generality, we assume `softmax` is employed for the output layer of the network and let $F(\cdot)_i$ denote the $i$-th dimension of the softmax output. When this DNN correctly classifies the input[1], *i.e.*, $C(x) = y$, where $y$ is the groundtruth label of the input $x$, our objective is to find an adversarial example $x'$ for $x$ such that they are imperceptibly close and yet the DNN classifier labels them distinctly; in other words, $C(x') \neq y$. In this paper, we bound the $\ell_p$ distance between an input and its adversarial counterpart: $\|x - x'\|_p < \tau_p, p = 2$ or $\infty$.

Different adversarial attack methods give rise to distinct adversarial examples for the same input, implying that the adversarial population for any input is at least greater than one. We impose a Gaussian distribution over the adversarial population. We then use the following process to generate an adversarial example from the population. Given an input $x$,

1. draw $z \sim \mathcal{N}(\theta(x), \sigma^2 I)$,

2. transform $\delta = \frac{1}{2}(\tanh(z) + 1) - x$,

3. clip $\delta' = \text{CLIP}_p(\delta)$, $p = 2$ or $\infty$, and

4. return $x' = x + \delta'$, an adversarial example for the input $x$.

In other words, we first draw a "seed" $z$ from the adversarial population $\mathcal{N}(\theta(x), \sigma^2 I)$ whose mean $\theta(x) \in \mathbb{R}^D$ conditions on the input $x$ and the variance $\sigma^2$ is left out as a free parameter to tune. In the second line, we transform the seed to the same range as the input by $\frac{1}{2}(\tanh(z) + 1) \in [0, 1]$, and then compute the offset $\delta$ between the transformed vector and the input. The third line clips the offset depending on which $\ell_p$ norm we use to bound the adversarial example. Finally, the clipped offset $\delta'$ is added to the input and returned as an adversarial example $x'$ for the input $x$.

The clip functions ensure that the final adversarial example is close enough to the input. While various forms of clipping are feasible, we use the following clip functions in this paper,

$$\text{CLIP}_2(\delta) = \begin{cases} \delta\tau_2/\|\delta\|_2 & \text{if } \|\delta\|_2 > \tau_2 \\ \delta & \text{else} \end{cases} \tag{1}$$

$$\text{CLIP}_\infty(\delta) = \min(\delta, \tau_\infty) \tag{2}$$

where the thresholds $\tau_2$ and $\tau_\infty$ are given by users.

Note that, in the above generation process, the Gaussian mean $\theta(x)$ is the only unknown to be estimated. We present an evolution strategy below to estimate this unknown. For simplicity, we drop $x$ out of the mean $\theta(x)$ in the rest of the paper, but beware that the adversarial Gaussian $\mathcal{N}(\theta, \sigma^2 I)$ is learned for a particular input $x$.

### 2.1    GAUSSIAN BLACK-BOX ADVERSARIAL ATTACK ($\mathcal{N}$ATTACK)

Recall that we want to fool DNN $F(\cdot)$ by an adversarial example $x'$, such that $y \neq \arg\max_i F(x')_i$ and $\|x' - x\|_p < \tau_p$. Carlini & Wagner (2017) have investigated a series of loss functions to capture this notion. Their experimental results show that the hinge loss, among a few others, behaves better

---

[1]We exclude the inputs for which the DNN predicts wrong labels in the experiments, following the convention of previous work (Carlini & Wagner, 2017).

---

**Algorithm 1** Gaussian black-box adversarial attack ($\mathcal{N}$ATTACK)

---

**Input:** DNN classifier $F(\cdot)$, input $x$ and its label $y$, initial mean $\theta_0$, standard deviation $\sigma$, learning rate $\eta$, sample size $N$, and the maximum number of iterations $T$
**Output:** $\theta_T$, mean of the adversarial population
 1: **for** $t = 0, 1, ..., T-1$ **do**
 2:     Sample $\epsilon_1, ..., \epsilon_N \sim \mathcal{N}(0, I)$
 3:     Compute losses $J_i := J(\theta_t + \sigma\epsilon_i)$ for $i = 1, \cdots, N$
 4:     Set $\theta_{t+1} \leftarrow \theta_t - \frac{\eta}{N\sigma} \sum_{i=1}^{N} J_i\epsilon_i$
 5: **end for**

---

than the straightforward cross-entropy loss. In this work, we also use the hinge loss defined as[2],

$$J(z) := \left[ \log F(x')_y - \max_{i \neq y} \log F(x')_i \right]^+ \tag{3}$$

where $[a]^+$ is short-hand for $\max(a, 0)$, $x'$ is determined by $z$ due to the adversarial generation process above, and $y$ is the groundtruth label of the input $x$.

Instead of focusing on any single adversarial example, we minimize the expected loss of the whole adversarial population, *i.e.*, $\min_\theta \mathbb{E}_{z \sim \theta} J(z)$, where the expectation is over $z$ drawn from the Gaussian $\mathcal{N}(\theta, \sigma^2 I)$. Once we find a good local optimum for the mean $\theta$ by solving $\min_\theta \mathbb{E}_{z \sim \theta} J(z)$, we can virtually generate an infinite number of adversarial examples to attack the DNN classifier.

**Optimization.**    We solve the problem $\min_\theta \mathbb{E}_{z \sim \theta} J(z)$ using an evolution strategy (Wierstra et al., 2008). It is a gradient search algorithm in a similar fashion as REINFORCE (Williams, 1992),

$$\nabla_\theta \mathbb{E}_{z \sim \theta} J(z) = \mathbb{E}_{z \sim \theta} \left[ J(z) \nabla_\theta \log P_\theta(z) \right] \tag{4}$$

$$\propto \mathbb{E}_{\epsilon \sim \mathcal{N}(0, I)} \left[ \epsilon / \sigma J(\theta + \sigma\epsilon) \right] \approx \frac{1}{N\sigma} \sum_{i=1}^{N} \left[ \epsilon_i J(\theta + \sigma\epsilon_i) \right], \tag{5}$$

where $P_\theta$ on the right-hand side of eq. (4) denotes the Gaussian distribution. Changing variable $z$ in eq. (4) by $\theta + \sigma\epsilon$, where $\epsilon$ follows a standard normal distribution, we arrive at eq. (5). This change of variable previously appears in (Salimans et al., 2017). Finally, we empirically estimate the gradient with respect to $\theta$ by using a standard normal sample of size $N$. Algorithm 1 summarizes the optimization procedure for the gradient search of the adversarial population's mean $\theta$.

### 2.2 REGRESSION BASED ADVERSARIAL ATTACK

The initialization to the mean $\theta_0$ plays a key role in the run time of Algorithm 1. When a good initialization is given, we often successfully find adversarial examples before reaching the maximal number of iterations $T$. Hence, we propose to accelerate the gradient search by using a regression neural network. It takes $x$ as the input and outputs $\theta_0$ to initialize Algorithm 1. In order to learn this regressor, we generate many input-adversarial-example pairs $\{(x, x')\}$ by running Algorithm 1 on the training set of benchmark datasets. The regression network's weights are then learned by minimizing the $\ell_2$ loss between the network's output and $\arctan(2x' - 1) - \arctan(2x - 1)$; in other words, we regress for the perturbation $\delta$ in the generation process above. Appendix A.4 presents more details about this regression network.

## 3 EXPERIMENTS

We use the proposed $\mathcal{N}$ATTACK to attack 11 defense methods for DNNs which are all published in 2018. For each defense method, we run experiments using the same protocol as reported in

---

[2]In this paper, we focus on untargeted attack, which is considered success once the DNN classifier predicts a wrong label for the adversarially perturbed input. Nonetheless, it is straightforward to revise the loss for dealing with targeted attack (Carlini & Wagner, 2017).

the original paper, including the datasets and $\ell_p$ distance (along with the threshold) to bound the difference from adversarial examples to inputs. In particular, CIFAR10 (Krizhevsky & Hinton, 2009) is employed in the attack on seven defense methods and ImageNet (Deng et al., 2009) is used for the remaining four. We examine all the test images of CIFAR10 and randomly choose 1,000 images from the test set of ImageNet. Nine of the defenses concern $\ell_\infty$ distance and one works with $\ell_2$ distance.

Our $\mathcal{N}$ATTACK achieves 100% attack success rate on six out of the 11 defense techniques and more than 90% on four (cf. Section 3.1). $\mathcal{N}$ATTACK, which is in the black-box attack realm, performs as strongly as or better than the white-box attack Backward Pass Differentiable Approximation (BPDA) (Athalye et al., 2018). It also significantly outperforms ZOO (Chen et al., 2017), a zero-th order gradient based black-box attack method as well as QL (Ilyas et al., 2018), a query-limited black-box attack based on evolution strategy.

Besides, we propose a new ROC curve to characterize the defense methods' effectiveness versus the strengths of the $\mathcal{N}$ATTACK attack (cf. Section 3.2). This curve provides a complementary metric to the attack success rate, uncovering more traits of the defense methods.

Finally, we observe relatively low transferabilities of the adversarial examples between the recently developed *defense techniques* (cf. Section 3.3). This is in sharp contrast to the findings that many adversarial examples are transferable across different vanilla *neural networks* trained on the same datasets. In some sense, this weakens the practical significance of white-box attack methods which could be applied to unknown DNN classifiers by attacking a substitute neural network instead (Papernot et al., 2017).

## 3.1 ATTACKING ELEVEN MOST RECENT DEFENSE TECHNIQUES

We consider 11 most recent defense techniques: Adversarial Training (ADV-TRAIN) Madry et al. (2018), Thermometer Encoding (THERM) (Buckman et al., 2018) , THERM-ADV (Athalye et al., 2018; Madry et al., 2018), Local Intrinsic Dimensionality (LID) (Ma et al., 2018), Cascade Adversarial Training (CAS-ADV) (Na et al., 2018), Stochastic Activation Pruning (SAP) (Dhillon et al., 2018), Randomization (Xie et al., 2018), Input Transformation (INPUT-TRANS) (Guo et al., 2018), Pixel Deflection (Prakash et al., 2018), Guided Denoiser (Liao et al., 2018), and Random Self-ensemble (RSE) (Liu et al., 2018).

To the best of our knowledge, our proposed $\mathcal{N}$ATTACK approach is the first black-box attack method that is able to achieve similar attack success rates as or even higher than the powerful white-box attack BPDA (Athalye et al., 2018), consistently on the above 11 defenses.

### 3.1.1 IMPLEMENTATION DETAILS

In our experiments, the defended DNNs of SAP, LID, RANDOMIZATION, INPUT-TRANS, THERM, and THERM-DAV come from (Athalye et al., 2018), the defended models of GUIDED DENOISER and PIXEL DEFLECTION are based on (Athalye & Carlini, 2018), and the models defended by RSE, CAS-ADV and ADV-TRAIN come from the original papers.

Following the previous practice of (Carlini & Wagner, 2017; Chen et al., 2017), we exclude the inputs mis-classified by the target model in (Athalye et al., 2018). The $l_\infty$ and $l_2$ distortion metrics are employed to bound the difference between an adversarial example and its corresponding clean input. We threshold the $l_\infty$ distance in the normalized $[0, 1]^D$ input space. The $l_2$ distance is the total root-mean-square distortion normalized by the number of pixels.

In all our experiments, we set $T = 600$ as the maximum number of optimization iterations, $N = 300$ for the sample size, variance of the isotropic Gaussian $\sigma^2 = 0.01$, and learning rate $\eta = 0.008$. $\mathcal{N}$ATTACK is able to defeat most of the defenses under this setting and about 90% inputs for other cases. We then fine-tune the learning rate $\eta$ and sample size $N$ for the hard leftovers.

The input dimension of CIFAR10 images is $32 \times 32 \times 3$. We directly search for an adversarial population mean $\theta$ of the same size by Algorithm 1. Whereas one may concern that the evolution search in such a high-dimensional space would not be feasible, we did not encounter any computational challenge for CIFAR10 images. However, for the ImageNet images whose sizes are $299 \times 299 \times 3$,

Table 1: Adversarial attack on 11 recently published defense methods. (** indicates the number reported in the original paper (Athalye et al., 2018). For all the other numbers, we obtain them by running the code and models released by the respective authors. BPDA, ZOO, QL and D-based stand for (Athalye et al., 2018), (Chen et al., 2017), (Ilyas et al., 2018) and (Brendel et al., 2017), respectively. * means the results are obtained on 1000 (200) randomly selected CIFAR10 (ImageNet) images, and the experiments on the full test set will be completed soon.) For D-based attack, we just report results on 100 images since it takes much longer time to converge only with hard-label and we will finish the experiments soon.

| Defense Technique | Dataset | Classification Accuracy % | Threshold & Distance | Attack Success Rate % | | | | |
|---|---|---|---|---|---|---|---|---|
| | | | | BPDA | ZOO | *QL | *D-based | $\mathcal{N}$ATTACK |
| *ADV-TRAIN (Madry et al., 2018) | CIFAR10 | 87.3 | 0.031 ($L_\infty$) | 46.9 | – | – | – | **47.9** |
| THERM-ADV (Athalye et al., 2018) | CIFAR10 | 88.5 | 0.031 ($L_\infty$) | 76.1 | 0.0 | 42.27 | – | **91.2** |
| LID (Ma et al., 2018) | CIFAR10 | 66.9 | 0.031 ($L_\infty$) | 95.0 | 92.9 | 95.73 | – | **100.0** |
| THERM (Buckman et al., 2018) | CIFAR10 | 92.8 | 0.031 ($L_\infty$) | **100.0** | 0.0 | 96.5 | – | **100.0** |
| SAP (Dhillon et al., 2018) | CIFAR10 | 93.3 | 0.031 ($L_\infty$) | **100.0** | 5.9 | 95.1 | – | **100.0** |
| RSE (Liu et al., 2018) | CIFAR10 | 91.4 | 0.031 ($L_\infty$) | – | – | – | – | **100.0** |
| CAS-ADV (Na et al., 2018) | CIFAR10 | 75.6 | 0.015 ($L_\infty$) | 85.0** | 96.1 | 68.37 | – | **97.7** |
| GUIDED DENOISER (Liao et al., 2018) | ImageNet | 79.1 | 0.031 ($L_\infty$) | **100.0** | – | – | – | 95.5 |
| RANDOMIZATION (Xie et al., 2018) | ImageNet | 77.8 | 0.031 ($L_\infty$) | **100.0** | 6.7 | 45.94 | – | 96.5 |
| INPUT-TRANS (Guo et al., 2018) | ImageNet | 77.6 | 0.05 ($L_2$) | **100.0** | 38.3 | 66.51 | 66.0 | **100.0** |
| PIXEL DEFLECTION (Prakash et al., 2018) | ImageNet | 69.1 | 0.015 ($L_\infty$) | 97.0 | – | 8.5 | – | **100.0** |

to speed up the execution, we first search for an adversarial population mean of the size $32 \times 32 \times 3$ and then up-sample it to the high resolution with bilinear interpolation.

### 3.1.2 ATTACK SUCCESS RATES

Table 1 shows the attack success rates of our black-box attack $\mathcal{N}$ATTACK, the white-box BPDA attack (Athalye et al., 2018), and the black-box ZOO attack (Chen et al., 2017). On 6 out of the 11 defense methods, $\mathcal{N}$ATTACK achieves 100% success rates. It also fails 4 defenses (THERM-ADV, CAS-ADV, GUIDED DENOISER, and RANDOMIZATION) for more than 90% of the tested inputs. And for ADV-TRAIN, we do succeed with some probability which is even better than white-box attack. We note the defense strengths of ADV-TRAIN, CAS-ADV and THERM-ADV come with price, *i.e.,* they lead to lower classification accuracies than the others on clean test images and it has been shown that adversarial retraining is difficult at ImageNet scale (Kurakin et al., 2016) — the third column of Table 1 shows the defended models' classification accuracies on the full test set of CIFAR10 and 1,000 randomly selected test images of ImageNet, respectively. At last but not the least, $\mathcal{N}$ATTACK is consistently on par with or better than the white-box attack BPDA, while ZOO fails to attack most of the defenses and QL can only get much lower success rate than $\mathcal{N}$ATTACK.

### 3.1.3 COMPARISON WITH QL

In this section, we compare our approach with Ilyas et al. (2018)'s in depth. As their focus was on query-limited (QL) black-box attack, we abbreviate their method as QL. Both $\mathcal{N}$ATTACK and QL employ the evolution strategy (ES) as the core optimization algorithm. While we use it to search for the adversarial Gaussian mean, QL uses it to estimate the gradients in PGD which is originally for white-box adversarial attack.

Table 2: Modifying QL (Ilyas et al., 2018) towards $\mathcal{N}$ATTACK step by step. We first add the `CLIP` operation to Line 3, Algorithm 2 (+`CLIP`), so that the input to the neural network is $\ell_\infty$ bounded. +loss replace QL's loss function with ours. $-$PGD removes the PGD step from Line 7, Algorithm 2, i.e., abandoning the clip and sign functions. +`tanh` is to lift the Gaussian distribution to the arctanh space of the adversarial perturbations (this changes Lines 3, 4, and 7 in Algorithm 2). +zscore is to subtract the mean from the losses and divide them by the standard deviation. We run the experiments with 1000 randomly selected images from the test set of CIFAR10.

| Targeted Model | QL | +CLIP | +loss | $-$PGD | +tanh | +zscore $\approx \mathcal{N}$ATTACK | $\mathcal{N}$ATTACK | QL+tanh +CLIP |
|---|---|---|---|---|---|---|---|---|
| THERM-ADV | 42.3 | 26.6 | 20.3 | 54.7 | 83.3 | 90.9 | **91.2** | 11.94 |
| SAP | 96.22 | 95.1 | 58.5 | 97.7 | 98.9 | 100 | **100** | 69.17 |

**Overall comparison.** We run QL using the same hyper-parameters as $\mathcal{N}$ATTACK for the ES part. As shown in Table 1, QL cannot perform on par with $\mathcal{N}$ATTACK on attacking LID, THERM, or SAP. Moreover, it leads to very low success rates on the defenses based on adversarial training (THERM-ADV and CAS-ADV). Finally, QL's performance on the defended neural networks for the ImageNet dataset (RANDOMIZATION, INPUT-TRANS, and PIXEL DEFLECTION) are also inferior to ours. We conjecture such results are mainly due to that, unlike the analysis shown before Section 2.1.2 in (Ilyas et al., 2018), ES is not an efficient estimator for the "true" gradients of PGD. The projection and sign operations in PGD violate the conditions of the analysis.

**Comparing QL with BPDA.** It is also worth comparing QL with BPDA because they employ PGD to generate $\ell_\infty$ bounded adversarial examples. More importantly, they both approximate the (obfuscated) gradients whereas in different ways. Table 1 shows the clear advantage of BPDA over QL, implying that ES is *not* as good as BPDA at estimating the true gradients.

**What makes $\mathcal{N}$ATTACK advantageous over QL?** Finally, we investigate the algorithmic differences between $\mathcal{N}$ATTACK and QL. To this end, we first describe the QL algorithm by using the same notations of this work (Algorithm 2).

---

**Algorithm 2** Query-limited black-box adversarial attack

**Input:** DNN classifier $F(\cdot)$, input $x$ and its label $y$, initial point $x_0$, standard deviation $\sigma$, learning rate $\eta$, sample size $N$, and the maximum number of iterations $T$
**Output:** $x_T$, an adversarial example
1: **for** $t = 0, 1, ..., T-1$ **do**
2:      Sample $\epsilon_1,...,\epsilon_N \sim \mathcal{N}(0, I)$
3:      Compute losses $F_i := \log F(x_t + \sigma \epsilon_i)_y$ for $i = 1, \cdots, N$
4:      Compute losses $F_{N+i} := \log F(x_t - \sigma \epsilon_i)_y$ for $i = 1, \cdots, N$
5:      Set $g \leftarrow \frac{1}{2N\sigma} \sum_{i=1}^{N} [F_i \epsilon_i - F_{N+i}\epsilon_i]$
6:      Decay learning rate $\eta \leftarrow \frac{\eta}{2}$ if $\log F(x_t)_y$ does not decrease continuously for 5 times
7:      $x_{t+1} \leftarrow \text{CLIP}_p(x_t - \eta \cdot \text{sign}(g) - x) + x$, $p = 2$ or $\infty$
8: **end for**

---

Comparing Algorithms 1 and 2, we hypothesize the following may have contributed to the differences in results (ranked by hypothesized importance),

**Line 3, Algorithm 1 *vs.* Line 3, Algorithm 2:** applying `CLIP`$_p$ to $x_t + \sigma\epsilon_i$ or not,

**Line 4, Algorithm 1 *vs.* Line 7, Algorithm 2:** using PGD or not,

**Line 3, Algorithm 1 *vs.* Line 3, Algorithm 2:** loss function,

**Line 3, Algorithm 1 *vs.* Line 3, Algorithm 2:** Gaussian sample or $\tanh(x)$ as the perturbation,

**Line 3, Algorithm 1 *vs.* Lines 4 & 5, Algorithm 2:** doubling the Gaussian sample or not.

In the same order as the above list, we gradually modify QL towards $\mathcal{N}$ATTACK and test each modified version on attacking SAP and THERM-ADV — the former is relatively weak and the latter

is very strong among the 11 defense techniques. Table 2 shows the results. It is obvious that there is a big performance jump for both SAP and THERM-ADV *after we remove PGD from QL, verifying the advantage of our approach for not depending on any white-box attack methods.* The second performance boost happens after we change the Gaussian distribution to the arctan space of the perturbation. While one may wonder this is not surprising considering that the arctan lifts the perturbation to the full real space which fits the Gaussian distribution better than the constrained adversarial perturbations, we argue that it is only effective after we also make the other changes to QL (see the last column of Table 2 for the inferior results of modifying QL only by the arctan). Finally, if we zscore the losses (subtract the mean and divide them by the standard deviation), the results approach $\mathcal{N}$ATTACK. In all the comparisons, we have kept both steps 3 and 4 in Algorithm 2, i.e., doubling the Gaussian sample for the modified versions of QL.

To conclude, PGD does play a key role in differentiating QL and $\mathcal{N}$ATTACK. In this sense, it is fair to say our work is fundamentally different from QL: QL is in the same vein as (Athalye et al., 2018) as they both rely on PGD, a white-box attack method, and both aim to approximate the gradients for PGD. In sharp contrast, we do not employ any white-box attack methods at all and, instead, provide a novel perspective to the adversarial attack by modeling the population of adversarial examples for every single image. This change alleviates the dependence on the gradients and leads to big differences in terms of the attack results.

## 3.2 A NEW ROC CURVE

The gradient search (Algorithm 1) has an appealing property; in expectation, the loss (eq. (3)) is reduced at every iteration. Despite there could be oscillations, we find that the attack strengths do grow monotonically with respect to the evolution iterations in our experiments. Hence, we propose a new ROC curve shown in Figure 1 featuring the attack success rate versus number of evolution iterations — strength of attack. For the experiment here, the Gaussian mean $\theta_0$ is initialized by $\theta_0 \sim \mathcal{N}(\arctan(2x-1), \sigma^2 I)$ for any input $x$ to maintain about the same starting points for all the curves.

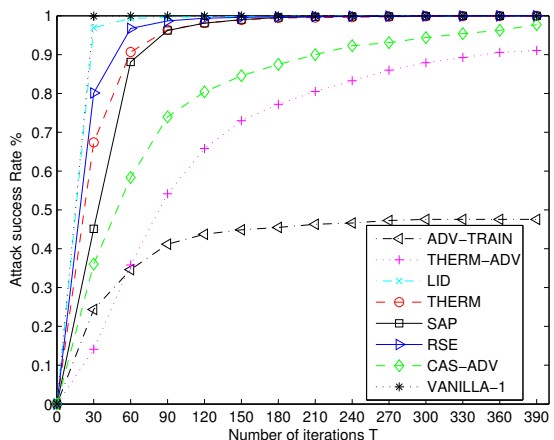

Figure 1: Success rate versus run steps of $\mathcal{N}$ATTACK.

Figure 1 compares seven defense methods on CIFAR10 along with a vanilla DNN classifier. It is clear that ADV-TRAIN, THERM-ADV and CAS-ADV are more difficult to attack than the others. What's more interesting is with the other four defenses and the vanilla DNN. Although $\mathcal{N}$ATTACK completely defeats all of them, the curve of the vanilla DNN is the steepest while the SAP curve rises much slower. If there are constraints on the computation time or the number of queries to the DNN classifiers, SAP becomes advantageous over RSE and THERM.

Note that the ranking of the defenses in Table 1 (*i.e.*, based on success rate) is different from the ordering in Figure 1, signifying the attack success rate and the ROC curve mutually complement each other. The curve reveals more characteristics of the defense methods especially when there are constraints on the computation time or number of queries to the DNN classifier.

## 3.3 TRANSFERABILITY

We also study the transferability of the adversarial examples across different defended DNNs. The confusion tables of BPDA and $\mathcal{N}$ATTACK are shown in Figure 2, respectively, where each number indicates the success rate of applying the adversarial examples originally targeting the row-wise defense model to attack the column-wise defenses. In addition to the defended DNNs, we also

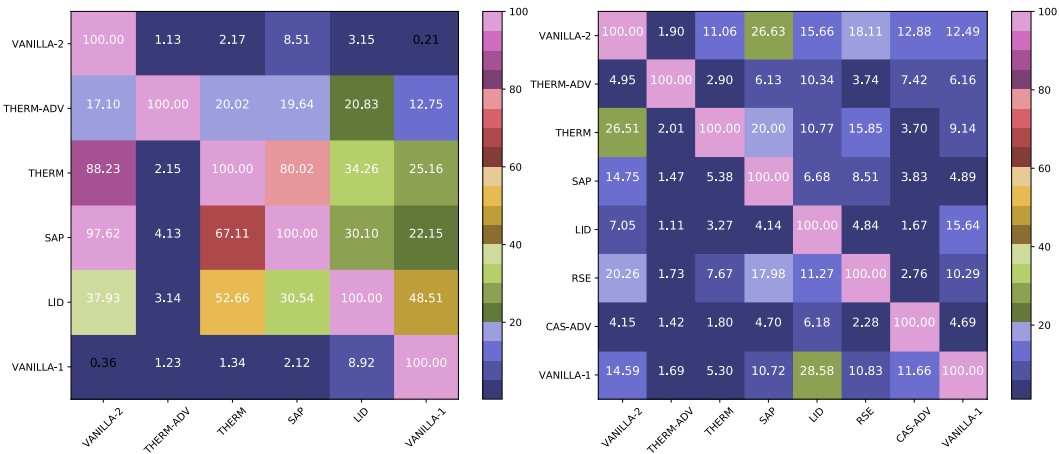

Figure 2: Transferabilities of BPDA (Athalye et al., 2018) (left) and $\mathcal{N}$ATTACK (right). We use the adversarial examples which target the defense of a row to attack all the column-wise defenses.

include two vanilla DNNs which are not equipped with any defense techniques: VANILLA-1 and VANILLA-2. VANILLA-1 is a light-weight DNN classifier built by Carlini & Wagner (2017) with 80% accuracy on CIFAR10. VANILLA-2 is the Wide-ResNet-28 (Zagoruyko & Komodakis, 2016) which gives rise to 92.3% classification accuracy on CIFAR10. For fair comparison, we change the threshold $\tau_\infty$ to 0.031 for CAS-ADV. We exclude RSE and CAS-ADV from the BPDA's confusion table because it is not obviously clear how to attack RSE using BPDA and the released BPDA code lacks the piece for attacking CAS-ADV.

Note that, unlike the experiments by Liu et al. (2016) about the adversarial examples' transferabilities across different vanilla DNNs, the experiments here are mainly concerned with the defended DNNs. Indeed, the results reveal some unique characteristics of the transferability under defense. There is an asymmetric pattern in the confusion tables; it is easier to transfer from defended models to the vanilla DNNs than vice versa. Besides, the overall transferrability is lower than that across the networks without any defenses (Liu et al., 2016). As below, we point out some extra observations.

First of all, the transferability of our black-box attack $\mathcal{N}$ATTACK is not as good as BPDA which is a white-box attack method. This is probably because BPDA is able to explore the intrinsically common part of the various DNN classifiers as it has the leverage over the true or estimated gradients that observe the DNNs' architectures and weights.

Secondly, both the network architecture and defense methods have an impact on the transferability. The VANILLA-2 network is the underlying classifier in SAP, THERM-ADV, and THERM. The adversarial examples originally attacking VANILLA-2 do transfer better to SAP and THERM than the others probably because they share the same DNN architecture, but they achieve very low success rate on THERM-ADV due to the defense technique.

Finally, it is worth noting that the transfer success rates are low no matter from THERM-ADV to the other defenses or the vice versa. Since all the other defended DNNs are trained following the empirical risk minimization principle while THERM-ADV employs the robust training (Madry et al., 2018), it is possible that the distinct transfer property of THERM-ADV attributes to the unique robust training method. We leave further exploration to the future work.

## 3.4 RUN TIME COMPARISON

Appendix A.2 studies the run time of our black-box attack $\mathcal{N}$ATTACK and the white-box attack BPDA. Results show that $\mathcal{N}$ATTACK is on par with BPDA on CIFAR10, both reaching an adversarial example in about 30s. To defeat an ImageNet image, it takes $\mathcal{N}$ATTACK about 71s without the regression network and 48s when it is equipped with the regression net; in contrast, BPDA only needs 4s. It is surprising to see that BPDA is almost 7 times faster at attacking a DNN on ImageNet than a DNN on CIFAR10, probably because the gradients of the former are not "obfuscated" as well as the latter.

## 4  RELATED WORK

Many approaches have been proposed to evaluate the robustness of DNNs as well as to attack and defend DNNs. Attacking approaches can be divided into two major categories, white-box attack and black-box attack. In contrast, defending neural network is a harder task and a few explorations have been exerted to improve the robustness of DNNs. We analyze the recent defenses in Appendix A.3.

**White-box attack.**  Preliminary studies on the robustness of DNNs focus on white-box setting with assuming full access to the targeted DNN. Szegedy et al. (2013) first prove DNN is fragile against adversarial examples and generate adversarial examples $x'$ similar to original sample $x$ in $\ell_2$ distance using box-constrained *L-BFGS*. Then the fast gradient sign (*FGS*) (Goodfellow et al., 2014a) method has been invented with two key differences from the *L-BFGS* method: first, it is optimized for $\ell_\infty$ distance metric, and secondly, it is designed primarily to be fast instead of producing very close adversarial examples. Papernot et al. (2016b) introduce an attack optimized under $l_0$ distance known as the Jacobian-based Saliency Map Attack (*JSMA*). *DeepFool* (Moosavi-Dezfooli et al., 2016) is an untargeted attack algorithm that aims to find the least $\ell_2$ distortion leading to misclassification by projecting an image to the closest separating hyperplane. Following these works, Carlini & Wagner (2017) propose an iterative optimization based attack (*C&W attack*), and then it seems to become a standard white-box attack approach. **Defense:** One common clue through those approaches is that they estimate sensitive regions of images by backward gradient to the pixels, and perturb them to attack the targeted DNNs.  *Obfuscated gradient* based defenses have been proposed to defeat gradient-based attacks like defensive distillation (Papernot et al., 2015) or most defenses listed in Appendix A.3 . Athalye et al. (2018) successfully attack those defenses by approximating gradients with BPDA.

**Black-box attack.**  The black-box attacking techniques do not exert the internal knowledge of DNN, and are more practical in the real applications. Thanks to the transferability property of adversarial examples (Szegedy et al., 2013), Papernot et al. (2017) can train a substitute DNN to imitate the behavior of the unknown DNN to be attacked, produce adversarial examples of the substitute, and then use them to attack. Chen et al. (2017) instead use zero-th order optimization to find adversarial examples. More recently, Brendel et al. (2017) introduce Boundary Attack, a decision-based attack that starts from a large adversarial perturbation and then seeks to reduce the perturbation while staying adversarial.

## 5  CONCLUSION

In this paper, we present an evolution gradient search based black-box attack approach, $\mathcal{N}$ATTACK, which is universally applicable and as powerful as the model-specific white-box approaches against most recent defenses. Extensive experiments show that $\mathcal{N}$ATTACK successfully attack 10 recently published defense techniques. Furthermore, a new ROC curve is proposed to analyze the robustness of defenses versus the strengths of adversarial attacks. Finally, we find that the defenses do weaken the transferability of the adversarial examples comparing to the transfer results across vanilla DNNs.

We make some remarks about the future work. Some existing works try to characterize the adversarial examples from the geometric view by pooling them of various inputs together. In contrast to this geometric and macro view, our work models the adversarial population of an input from a probabilistic and micro view. There are still a lot to explore along the avenue of explicitly modeling the adversarial population, as opposed to a single adversarial example, for any input to a DNN.

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

Table 3: Comparison of C&W attack, ZOO, and $\mathcal{N}$ATTACK on L2 distortion. '–' means C&W attack fails on Therm. *$\mathcal{N}$ATTACK uses the optimization shown in eq. (6).

| Targeted Model | C&W attack Carlini & Wagner (2017) | ZOO Chen et al. (2017) | *$\mathcal{N}$**ATTACK** | $\mathcal{N}$**ATTACK** |
|---|---|---|---|---|
| THERM ($L_2$) | – | 0.8868 | 0.26 | 0.35 |
| VANILLA-1 ($L_2$) | 0.17 | 0.19 | 0.25 | 0.33 |

# A  APPENDIX

## A.1  COMPARISON WITH STANDARD $l_2$ DISTORTION

We compare our approach with several adversarial attack baselines including Carlini & Wagner (2017) (C&W) and ZOO Chen et al. (2017) on a typical defense model THERM (Buckman et al., 2018) and the DNN VANILLA-1 in terms of $l_2$ distortion. The defense model THERM is based on the DNN VANILLA-2 on CIFAR10 trained with thermometer encoding. We run $\mathcal{N}$ATTACK with default setting, and with an additional trick to decrease $l_2$ distortion by using the objective shown in eq. (6) which is first introduced in C&W attack (Carlini & Wagner, 2017) (denoted by *$\mathcal{N}$ATTACK in Table 3). As shown in Table 3, $\mathcal{N}$ATTACK can get comparable lower distortion on both standard model and defense with 100% success rate while C&W attack fails on Therm, and ZOO gets very large distortion on Therm.

$$J_{c\&w}(\delta) = \|\delta\|_2 + c \cdot f(x + \delta) \quad f(x + \delta) := \left[ \log F(x + \delta)_y - \max_{i \neq y} \log F(x + \delta)_i \right]^+ \quad (6)$$

## A.2  COMPARISON WITH RUNNING TIME

Compared with the white-box attack approach BPDA (Athalye et al., 2018), $\mathcal{N}$ATTACK may take longer time since BPDA can find the local optimal solution quickly being guided by the approximate gradients. However, the evolution strategy based algorithms can be parallel when running each episode, as discussed in Salimans et al. (2017). We attack 100 samples on one machine with 4*TITAN-XP graphic cards and calculate the average running time of attacking. As shown in Table 4, $\mathcal{N}$ATTACK can get even faster attack than the white-box attack approach BPDA on CIFAR-10, yet performs far slower on ImageNet. The main reason is that when the image size is (3*32*32), the search space is tolerable; however, the running time could be lengthy for high resolution images like ImageNet examples (3*299*299) especially for some hard cases (nearly 90% images can be attacked in one minute but it could take about 60 minutes for some hard cases).

We adopt a regression FCN to approximate a good initialization of $\theta_0$ and we name $\mathcal{N}$ATTACK initialized with regression net as $\mathcal{N}$ATTACK-R. We run $\mathcal{N}$ATTACK and $\mathcal{N}$ATTACK-R on ImageNet with the population size $n = 40$. Thus the success rate for $\mathcal{N}$ATTACK with random initialization is 82% and for $\mathcal{N}$ATTACK-R is 91.9%, which proves the efficiency of regression net. The running time shown in Table 4 is calculated on the images with successful attacks. The results demonstrate that $\mathcal{N}$ATTACK-R can decrease 22.5s attacking time per image compared with the random initialization.

Table 4: Average attacking time for one image. $\mathcal{N}$**ATTACK-R** represents $\mathcal{N}$ATTACK initialized with regression net

| Defense | Dataset | BPDA Athalye et al. (2018) | $\mathcal{N}$ATTACK | $\mathcal{N}$**ATTACK-R** |
|---|---|---|---|---|
| SAP Dhillon et al. (2018) | CIFAR-10 ($L_\infty$) | 33.3s | 29.4s | – |
| RANDOMIZATION Xie et al. (2018) | ImageNet ($L_\infty$) | 3.51s | 70.77s | 48.22s |

### A.3 TEN MOST RECENT DEFENSE TECHNIQUES

This paper attacks 10 most recent defense techniques, as shown below.

- **Thermometer encoding (THERM).** To break the hypothesized linearity behavior of DNNs (Goodfellow et al., 2014a), Buckman et al. (2018) propose to transform the input by non-differentiable and non-linear thermometer encoding, followed by a slight change to the input layer of conventional DNNs.

- **ADV-TRAIN & THERM-ADV.** Madry *et al.* propose a defense using adversarial training (ADV-TRAIN). Specially, the training procedure alternates between seeking an "optimal" adversarial example by project gradient descent (PGD) and minimizing the classification loss under the PGD attack. Furthermore, Athalye et al. (2018) find that the adversarial robust training Madry et al. (2018) can significantly improve the defense strength of THERM (THERM-ADV). Compare with ADV-TRAIN, the adversarial examples are produced by Logit-Space Projected Gradient Ascent (LS-PGA) during the training process.

- **Local intrinsic dimensionality (LID).** Ma et al. (2018) propose a metric to capture the local intrinsic dimension of the inputs and adversarial examples. This metric is shown to be effective in distinguishing adversarial and clean images.

- **Cascade adversarial training (CAS-ADV).** Na et al. (2018) improve the adversarial training (Goodfellow et al., 2014b; Kurakin et al., 2016) in a cascade manner. A model is trained from the clean data and one-step adversarial examples first. The second model is trained from the original data, one-step adversarial examples, as well as iterative adversarial examples generated against the first model. Additionally, a regularization is introduced to the unified embeddings of the clean and adversarial examples.

- **Stochastic activation pruning (SAP).** Dhillon et al. (2018) randomly drop some neurons of each layer with the probabilities in proportion to their absolute values.

- **RANDOMIZATION.** Xie et al. (2018) add a randomization layer between inputs and a DNN classifier. This layer consists of resizing an image to a random resolution, zero-padding, and randomly selecting one from many resulting images as the actual input to the classifier.

- **Input transformation (INPUT-TRANS).** By a similar idea as above, Guo et al. (2018) explore several combinations of input transformations coupled with adversarial training, such as image cropping and rescaling, bit-depth reduction, JPEG compression.

- **PIXEL DEFLECTION.** Prakash et al. (2018) randomly sample a pixel from an image and then replace it with another pixel randomly sampled from the former's neighborhood. Discrete wavelet transform is also employed to filter out adversarial perturbations to the input.

- **GUIDED DENOISER.** Liao et al. (2018) use a denoising network architecture to estimate the additive adversarial perturbation to an input.

- **Random self-ensemble (RSE).** Liu et al. (2018) combine the ideas of randomness and ensemble using the same underlying neural network. Given an input, it generates an ensemble of predictions by adding distinct noises to the network multiple times.

### A.4 DETAILS OF THE REGRESSION NETWORK

Concerning initialization of $\theta$, the naive way is $\theta_0 = \arctan(2x - 1) + \phi$ where $\phi$ is sampled from Gaussian distribution $\mathcal{N}(0, \sigma^2 I)$. An observation is that we can get adversarial perturbation in a short time with smaller population size $n$ for most examples. As a result, we can get plenty of $(x, \phi)$ pairs in a short time. Those phenomenons inspire us to train a segmentation model $S$ to regress the perturbation of input images. A simple FCN (Shelhamer et al., 2016) model pretrained on PASCAL VOC segmentation challenging (Everingham et al., 2010) is used here. Furthermore, we change the last two convolutional layers to ensure the output is (3*32*32) and replace the loss function with mean square loss to regress the perturbation. At the test time, given an unseen image x, the output of segmentation model can be treated as the initialization of $\phi$, *i.e.*, $\phi_0 = S(x)$, and thus we can accelerate the attack process. The effectiveness of this approach is discussed in Appendix A.2.

