# OpenReview forum: "NATTACK: A STRONG AND UNIVERSAL GAUSSIAN BLACK-BOX ADVERSARIAL ATTACK"
_ICLR.cc/2019/Conference_

### Official Review · AnonReviewer3 · 2018-10-31
**Good evaluation but important prior work was missed which substantially reduces novelty and makes a major rewrite necessary**

**Rating:** 4
**Confidence:** 5

**Review:**

In this work the authors use a score-based adversarial attack (based on the natural evolution strategy (NES)) to successfully attack a multitude of defended networks, with success rates rivalling the best gradient-based attacks.

As confirmed by the authors in a detailed and very open response to a question of mine, the attack introduced here is actually equivalent to [1]. While the attack itself is not novel (which will require a major revision of the manuscript), the authors point out the following contributions over [1]:

* Attack experiments here go way beyond Ilyas et al. in terms of Lp metrics, different defense models, different datasets and transferability.
* Different motivation/derivation of NES.
* Concept of adversarial distributions.
* Regression network for good initialization.
* Introduction of accuracy-iterations plots.

My main concerns are as follows:
* The review of the prior literature, in particular on score-based and decision-based defences (the latter of which are not even mentioned), is very limited and is framed wrongly. In particular, the statement “However, existing black-box attacks are weaker than their white-box counterparts” is simply not true: as an example, the most prominent decision-based attack [2] rivals white-box attacks on vanilla DNNs as well as defended networks [3].
* The concept of adversarial distributions is not new but is common in the literature of real-world adversarials that are robust to transformations and perturbations (like gaussian noise), check for example [4]. In [4] the concept of _Expectation Over Transformation (EOT)_ is introduced, which is basically the generalised concept of the expectation over gaussian perturbations introduced in this work.
* While I like the idea of accuracy-iterations plots, the idea is not new, see e.g. the accuracy-iterations plot in [2] (sample-based, Figure 6), the loss-iterations plot in [5] or the accuracy-distortion plots in [3]. However, I agree that these type of visualisation or metric is not as widespread as it should be.

Hence, in summary the main contribution of the paper is the application of NES against different defence models, datasets and Lp metrics as well as the use of a regression network for initialisation. Along this second point it would be great if the authors would be able to demonstrate substantial gains in the accuracy-query metric. In any case, in the light of previous literature a major revision of the manuscript will be necessary.

[1] Ilyas et al. (2018) “Black-box Adversarial Attacks with Limited Queries and Information” (https://arxiv.org/abs/1804.08598)
[2] Brendel et al. (2018) “Decision-Based Adversarial Attacks: Reliable Attacks Against Black-Box Machine Learning Models” (https://arxiv.org/abs/1712.04248)
[3] Schott et al. (2018) “Towards the first adversarially robust neural network model on MNIST” (https://arxiv.org/abs/1805.09190)
[4] Athalye et al. (2017) “Synthesizing Robust Adversarial Examples” (https://arxiv.org/pdf/1707.07397.pdf)
[5] Madry et al. {2017) “Towards Deep Learning Models Resistant to Adversarial Attacks” (https://arxiv.org/pdf/1706.06083.pdf)

---

> ### Author Response · Authors · 2018-11-23
> **PDF updated & responses to questions**
>
> Q: the attack introduced here is actually equivalent to (Ilyas et al., 2018).
>
> A: We believe the above is a mis-interpretation of our responses. Denote by QL (Ilyas et al., 2018). QL is hinged on the white-box PGD attack --- in terms of the methodology, it is actually closer to [1] than ours because both QL and [1] essentially approximate the gradients for PGD. As a result, the quality of the estimated gradients in QL is a big deal. Unfortunately, ES does not give rise to stable gradients due to the sampling step and PGD’s projection and sign functions. Indeed, after we remove the PGD step in QL, there is a significant performance boost (cf. Table 2 in the revised PDF). On the contrary, we do not employ any white-box attack methods at all in developing our algorithm. The Gaussian mean is more important than the gradients in our approach. Whereas ES is a natural choice to search for the Gaussian mean, some derivative-free methods [2] are also good alternatives.
>
> Please see Section 3.1.3 for a more detailed investigation about QL.
>
>
> Q: "existing black-box attacks are weaker than their white-box counterparts” is simply not true.
>
> A: It is actually unclear how strong the existing black-box methods are on attacking the defended neural networks --- most experiments reported in the original publications are conducted on vanilla neural networks. Our own experiments do show that ZOO [1] and the decision-based attack [2] fail to perform well on attacking all the 10 defense methods (cf. Table 1 in the PDF) --- as the decision-based attack consumes a lot of run time, we have to include the complete results later. We have toned down the description about prior black-box attack in the introduction (cf. the text highlighted in the blue color).
>
> [1] Chen, Pin-Yu, et al. "Zoo: Zeroth order optimization based black-box attacks to deep neural networks without training substitute models." Proceedings of the 10th ACM Workshop on Artificial Intelligence and Security. ACM, 2017.
>
> [2] Wieland Brendel, Jonas Rauber, and Matthias Bethge. Decision-based adversarial attacks: Reliable attacks against black-box machine learning models. arXiv preprint arXiv:1712.04248, 2017.
>
>
> Q: The concept of adversarial distributions is not new
>
> A: We have to point out the distribution over adversarial examples per image is different from the distribution over the transformations for a physical adversarial in the real world. In order to photograph a real-world adversarial, it is natural to consider all the conditions (location, background, lighting, etc.) as a distribution of transformations. In contrast, it is not so obvious to model by a distribution the adversarial examples for every single image. To the best of our knowledge, this work is the first to capture the whole population of adversarial examples per image.

---

> > ### Comment · AnonReviewer3 · 2018-11-25
> > **The "sharp contrast" you are trying to construct does not exist**
> >
> > Dear authors,
> > first of all let me say that I do appreciate the effort you put into revising the manuscript and the rebuttal. I have, however, a couple of questionmarks behind your responses:
> >
> > 1) Bad performance of decision- and score-based attacks
> >
> > First, [2] is an L2-based attack but you are applying it in an L-infinity scenario, that doesn't make sense. Second, the decision-based attack [2] and other black-box attacks based on score-based methods do perform very well on e.g. Madry et al. (MNIST) or the analysis by synthesis model [3], see results in [3]. In fact, on Madry et al. [2] performs much better than e.g. gradient-based BIM. I hence strongly doubt the results related to [2].
> >
> > 2) "We run QL using the same hyper-parameters as N ATTACK for the ES part"
> >
> > That's not a fair comparison because the optimal parameters for QL are likely different (e.g. because of the clipping) than for the N Attack. Please compare the attacks with hyper parameters optimised for each attack.
> >
> > 3) On the contrary, we do not employ any white-box attack methods at all in developing our algorithm.
> >
> > I think you are confusing what is meant by "white-box": white-box refers to whether or not you are using the backpropagated gradient (which requires you to have access to the internal structure and weights of the model). PGD is white-box if you use the exact gradient and score-based if you use estimated gradients. Similarly, your method is performing a gradient descent using an estimated gradient. I fail to see how that fundamentally differs from QL.
> >
> > 3) "this work is the first to capture the whole population of adversarial examples per image"
> >
> > You are not capturing the whole distribution, you are capturing a local Gaussian region. I really do not like the whole wording around "populations" and find that confusing and misleading. Your motivation in the end is exactly equivalent to ES, you are just using the gradient in a different way.
> >
> > 4) In the updated manuscript you write: "In sharp contrast, we do not employ any white-box attack methods at all and, instead, provide a novel perspective to the adversarial attack by modeling the whole population of adversarial examples for every single image. This change alleviates the dependence on the gradients and leads to big differences in terms of the attack results."
> >
> > In line with what I wrote above I find this part extremely misleading. Of course your method is based on a "white-box attack method" - it's called gradient descent. There is no sharp contrast, just as there is no sharp contrast between BIM and MIM, both are based on a similar principle. I think it would be much better if you would tune your paper to say that you have developed a more effective ES-based attack and show that an PGD-based ES attack doesn't work as well on defended networks.

---

> > > ### Author Response · Authors · 2018-11-27
> > > **Regarding the experiments**
> > >
> > > We have tuned the hyper-parameters of the competing methods (BPDA, QL, ZOO, D-based) in order to achieve the best performances they could have. The ES part is to approximate an expectation by a sample mean, so we believe it is fair to fix the sample size for QL and our algorithm --- we did tune the other hyper-parameters in QL such as the learning rate and number of iterations. (QL actually doubles the sample size by reversing the signs of the samples.)
> > >
> > > For all of the competing methods but the decision-based, we used the code released by the original authors in the experiments. We did not find the official implementation of the decision-based method (D-based) due to the deadline rush; instead, this implementation (https://github.com/greentfrapp/boundary-attack) was employed. Thanks to the reviewer's question, we tested this implementation using the evaluation metric reported in the original publication and only found it failed to re-produce the reported results. Upon a second search, we found the "foolbox" implementation (https://github.com/bethgelab/foolbox) of D-based. With it, we re-produced the reported results and obtained 66% success rate on attacking INPUT-TRANS (ours: 100%, BPDA: 100%, QL: 66.5%, and ZOO: 38.3%). Regarding the D-based experiments for generating the $\ell_infty$ bounded adversarial examples, we re-wrote the norm function in the two implementations and did not observe any good results. We agree with the reviewer that, since D-based was particularly tailored for the $\ell_2$ bounded adversarial examples, that simple change of norm is not good enough to improve D-based for handling the $\ell_infty$ metric. More careful work has to be done to modify the D-based method to fit the $\ell_infty$ context; for example, the projection to the $\ell_2$ sphere has to be updated by the projection to the $\ell_infty$ polygon. We have updated the PDF.

---

> > > ### Author Response · Authors · 2018-11-27
> > > **Understanding our approach**
> > >
> > > Regarding the "whole" wording, we are fine to remove the "whole" because
> > > "the whole population of adversarial examples per image"
> > > is actually equivalent to
> > > "the population of adversarial examples per image".
> > > Given an image, all its adversarial examples comprise the population. We use a Gaussian distribution to model this population in this work. In the future, other multi-variate continuous distributions, like GMM or uniform, may be found a better fit to the population. Additionally, one may also consider to capture the population by non-parametric distributions. No matter which one --- including the Gaussian, what it models is the population per image and not the local region.
> > >
> > > Thanks to the above, we formalize our problem as minimizing the expected loss under the Gaussian distribution. This problem formulation is different from any problem formulations of the existing white-box attack methods. In contrast, BPDA and QL are built upon PGD (and CW for BPDA). PGD is the basic framework for them and they only (and yet non-trivially) replace the true gradients by the estimated ones. In this sense, we said we did not employ any white-box attack methods. We also agree with the reviewer that there is no sharp contrast between BIM and MIM because both are based on the similar principle of solving the following optimization problem: $min_{perturbation} Loss$. In contrast, ours is $min_{Gaussian mean} Expectation Loss$.

---

> > > > ### Comment · AnonReviewer3 · 2018-12-04
> > > > **you don't want to model the distribution of adversarials**
> > > >
> > > > You don't want to model the distribution of adversarial examples. Say you use a more powerful distribution that is really capable of capturing the distribution of adversarial examples. This distribution would comprise the whole input space for which the model decision is different from the label of the original image, right? You would then take the mean of that distribution (at least that's what you do right now). But that point would certainly be far away from the minimum adversarial perturbation that you seek.

---

> > > > > ### Author Response · Authors · 2018-12-08
> > > > > **Mode is not the mean**
> > > > >
> > > > > If you referred to a parametric distribution by the "more powerful distribution", we learn the parameters to specify the distribution as we described earlier. Once we reach such a distribution, we are supposed to either sample from it or use the mode to generate an adversarial example following Steps 1--4 in the paper. The mean of Gaussian overlaps with the Gaussian's mode.

---

> > > > > > ### Comment · AnonReviewer3 · 2018-12-08
> > > > > > **minimal adversarial examples?**
> > > > > >
> > > > > > Sampling from this distribution would have little to do with minimal adversarial examples: if you succeeded in modelling all adversarials, this would include the vast majority that are far away from the clean image. Sampling would mainly return large adversarial perturbations, not small ones.

---

> > > > > > > ### Author Response · Authors · 2018-12-08
> > > > > > > **Steps 1--4 generate valid adversarial examples**
> > > > > > >
> > > > > > > Not sure why you are obsessed with the minimal adversarial examples. If those are what you are looking for, our paper does not provide a direct answer though you probably can derive one based on our work.
> > > > > > >
> > > > > > > Kindly check Steps 1--4 in the paper which generate valid adversarial examples whose differences from the original images are imperceptible up to the thresholds $\tao_p, p=2 or \infty$.

---

### Official Review · AnonReviewer2 · 2018-11-01
**NATTACK: A STRONG AND UNIVERSAL GAUSSIAN BLACK-BOX ADVERSARIAL ATTACK**

**Rating:** 4
**Confidence:** 3

**Review:**

Summary: In this paper the authors discuss a black-box method to learn
adversarial inputs to DNNs which are "close" to some nominal example
but nevertheless get misclassified. The algorithm essentially tries to
learn the mean of a joint Gaussian distribution over image
perturbations so that the perturbed image has high likelihood of being
misclassified. The method takes the form of zero-th order gradient
updates on an objective measuring to what degree the perturbed example
is misclassified. The authors test their method against 10 recent DNN
defense mechanisms, which showed higher attack-success rates than
other methods. Additionally the authors looked at transferrability of
the learned adversarial examples.

Feedback: As noted before, this paper shares many similarities with

[1] "Black-box Adversarial Attacks with Limited Queries and Information" (https://arxiv.org/abs/1804.08598)

and the authors have responded to those similarities in two follow-ups. I have reviewed these results and their
method does appear to improve over [1]. However, I am still reluctant to admit these additions to the original submission,
mainly because dropping [1] in the original submission seems to be a fairly major omission of one of the most relevant competitors out there. In its current form, the apparent redundancies distract significantly from the paper, and to remedy this, the paper would have to change significantly in order to relate it properly to [1] clear is needed. I'd be curious on the ACs thoughts on this.

I appreciate the authors' claim that their method can breach many of the popular defense methods out there, but we
also see that many of the percentages in Figure 1 converge  to 1. On the one hand this suggests that all defense methods
are in some sense equally bad, but on the other, it could also just reflect on the fact that the thresholds are chosen
"too large". I understand that many of the thresholds were inherited from previous work, but it would nevertheless help if the authors showed some example adversarial images to help baseline this Figure.

---

> ### Author Response · Authors · 2018-11-23
> **PDF revised with extensive discussion on [1]; Curves updated**
>
> Q: The paper would have to change significantly in order to relate it properly to (Ilyas et al., 2018).
>
> A: Denote by QL (Ilyas et al., 2018)’s approach. We have added to the revised PDF
> + a new paragraph (in the Introduction section) to draw readers’ attention to QL upfront,
> + new results of QL on attacking the 10 defense methods (Table 1), and
> + a new section (Section 3.1.3) to carefully investigate the factors that contribute to the inferior performance of QL algorithm. The results reveal that, in order to improve its attack success rates, it is vital to get rid of PGD (projection and the sign of the gradients), which is the foundation upon which QL is built, and meanwhile to couple the $\ell_infty$ clip with the tanh transformation.
>
> Given the above changes, it seems like feasible to extensively discuss QL and yet not completely re-write the paper.
>
> Additionally, we wanted to emphasize that the Gaussian mean is more important than the gradients in our approach. Whereas ES is a natural choice to search for the Gaussian mean, some derivative-free methods [2] are also good afternatives. In sharp contrast, QL is hinged on the white-box PGD attack --- in terms of the methodology, it is actually closer to [1] than ours because both QL and [1] essentially approximate the gradients for PGD. As a result, the quality of the estimated gradients in QL is a big deal. Unfortunately, ES does not give rise to stable gradients due to the sampling step and PGD’s projection and sign functions. Indeed, after we remove the PGD step in QL, there is a significant performance boost (cf. Table 2).
>
> [1]	Anish Athalye, Nicholas Carlini, and David Wagner. Obfuscated gradients give a false sense of security: Circumventing defenses to adversarial examples. arXiv preprint arXiv:1802.00420, 2018.
> [2] Luis Miguel Rios and Nikolaos V Sahinidis. Derivative-free optimization:  a review of algorithms and comparison of software implementations. Journal of Global Optimization, 56(3):1247–1293,2013.
>
>
> Q: Example adversarial examples to baseline the figure:
>
> A: Sorry for the confusion about Figure 1. First of all, we did not include all the defense methods in Figure 1 due to the heavy run time on ImageNet. Besides, for each attack method, we had removed all the examples of which it failed to change the labels. Our intention was to compare the relative convergences when their last steps are aligned. Upon reading your comments, however, we think this alignment is actually unnecessary and should be removed. In the revised PDF submission, you can see that some of the attack methods fail to reach 100% success rate.
>
> We will add some example adversarial examples in the appendix, but the adversarial examples in $\ell_\infty = 0.031$ are hardly differentiable from the benign ones.

---

### Official Review · AnonReviewer1 · 2018-11-05
**original**

**Rating:** 7
**Confidence:** 3

**Review:**

In this paper, authors propose a "universal" Gaussian balck-box adversarial attack.
Original and well-written (although there are a few grammar mistakes that would require some revision) and structured. Having followed the comments and discussion I am convinced that the proposed method is state of the art and interesting enough fro ICLR.
To the best of my knowledge, the study is technically sound.
It fairly accounts for recent literature in the field.
Experiments are convincing.
One thing I am not so convinced about is the naming of the evaluation curve as "a new ROC curve". I understand the appeal of pairing the proposed evaluation curve with the ROC curve but, beyond an arguable resemblance, they have no much in common, really.

---

> ### Author Response · Authors · 2018-11-23
> **Appreciated; Have re-named the curve**
>
> Thank you for the encouraging comments! Regarding the name of the curve, we have removed “ROC” and now simply call it the curve of success rate vs. number of evolution iterations. We will continue to polish the text.

---

### Public Comment · (anonymous) · 2018-10-09
**Code released: https://github.com/gaussian-attack/Nattack**

Code released: https://github.com/gaussian-attack/Nattack

---

### Comment · AnonReviewer3 · 2018-10-23
**Difference to state of the art**

Could the authors elaborate as to how this attack differs from [1]? As far as I can see this work uses the same gradient estimate with Gaussian bases.

[1] "Black-box Adversarial Attacks with Limited Queries and Information" (https://arxiv.org/abs/1804.08598)

---

### Comment · AnonReviewer3 · 2018-10-23
**Comparison with state-of-the-art**

Could the authors elaborate as to how this attack differs from [1]? As far as I can see this work uses the same gradient estimate with Gaussian bases.

[1] "Black-box Adversarial Attacks with Limited Queries and Information" (https://arxiv.org/abs/1804.08598)

---

> ### Author Response · Authors · 2018-10-24
> **Differences from "Black-box Adversarial Attacks with Limited Queries and Information"**
>
> That’s a great catch. Thank you very much! We should have read the paper before… It is intriguing (and yet disappointing for us) to see that a similar approach has been proposed (Ilyas et al. 2018) by also resorting to the natural evolution strategy (NES), but it is not surprising. After all, derivative-free methods, such as NES, REINFORCE, and the zero-th order algorithms, are a natural choice for the blackbox attack.
>
> While we mainly attack up to 10 recently published defense methods by the proposed approach, Ilyas et al. (2018) focus on attacking a vanilla neural network under the constraints of limited queries and information (e.g., top k entries as opposed to the full output vector).
>
> On the algorithmic aspect, both ours and Ilyas et al. (2018)’s employ NES as the optimization algorithm. However, we arrive at it via different routes and for different purposes. We assume a probabilistic generation process of the adversarial examples (Steps 1–4, Section 2), which finds an adversarial example by a one-step addition to the input. In contrast, Ilyas et al. (2018)’s modeling assumption is that an adversarial example can be found by PGD, which iteratively updates the original input with a small learning rate until it becomes adversarial. To this end, we use NES to estimate the parameters of the distribution, while Ilyas et al. (2018) use NES to replace the true (stochastic) gradients in PGD. We contend that, due to the non-differentiable clip and projection operations and the fairly large Gaussian covariance, NES is *not* an efficient (and possibly a biased) estimator of the true gradients — we are running experiments to empirically verify if this is true or not.
>
> It is a conceptual change from the traditional attack methods (e.g., PGD) to the way of modeling the adversarial examples by a distribution. This change may enable some exciting future works. For instance, we can draw samples from the distribution to characterize the adversarial boundaries, efficiently do adversarial training, etc.
>
> Another notable difference from (Ilyas et al. 2018)’s is that we train a regression neural network to find a good initialization for NES. Experiments verify the benefit of this regression network.
>
> On the experimental aspect, we attack the recently proposed defense methods following the protocols set up in the original papers. As a result, we experiment with both CIFAR10 and ImageNet, both the $\ell_2$ and $\ell_infty$ distances, and different types of defenses (e.g., input randomization and discretization, ensembeling, denoising, etc.). In contrast, Ilyas et al. (2018) experiment with ImageNet with an $\ell_\infty$ distance. In addition, we examine the adversarial examples’ transferabilities across different defense methods. Unlike the findings about the transferability across vanilla neural networks, our results indicate several unique characteristics of the transferability of our adversarial examples for the defended neural networks (cf. Section 3.3). Finally, we plot the curves of the attack success rates versus the iteration numbers, a new evaluation scheme which is complementary to the final attack success rates.

---

> > ### Author Response · Authors · 2018-11-01
> > **Experimental comparison with "Black-box Adversarial Attacks with Limited Queries and Information"**
> >
> > With the open-source code released by Ilyas et al. (2018), we have evaluated their method on attacking three defense methods: SAP and Therm for CIFAR10 and Randomization for ImageNet. The results (success rate vs. number of optimization iterations) are shown in the tables below. We have also tested larger sample size and higher number of iterations for NES, and yet the results remain about the same.
> >
> > Ilyas A, Engstrom L, Athalye A, Lin J. Black-box Adversarial Attacks with Limited Queries and Information. arXiv preprint arXiv:1804.08598. 2018 Apr 23.
> >
> > The inferior attacking results of (Ilyas et al., 2018) verify our conjecture above, i.e., due to the non-differentiable clip and projection operations and the fairly large Gaussian covariance, NES is *not* an efficient (and possibly a biased) estimator of the true gradients of PGD. As a result, NES is not able to approach PGD’s strong attack  performance.
> >
> > Table 1: Success rate on attacking SAP (CIFAR10)
> > # of iterations           30       90       150      210       270     300      360      400
> > Ours                          45.13  96.21  99.00   99.54   99.81   100      100      100
> > Ilyas et al. (2018)'s  33.36  34.51  36.03   37.36   37.36   37.36   37.36   37.36
> >
> > Table 2: Success rate on attacking Therm (CIFAR10)
> > # of iterations            30       90       150      210       270     300       360     400
> > Ours                          67.38   96.38  98.92   99.53   99.74   99.89   100     100
> > Ilyas et al. (2018)'s  59.22   83.32  83.82   84.32   85.33   85.33   85.33   85.33
> >
> > Table 3: Success rate on attacking Randomization (ImageNet)
> > # of iterations          30       90       150      210       270    300    360    400
> > Ours                         21.54  78.58  90.02   95.41   95.5    95.5   95.5   95.5
> > Ilyas et al. (2018)'s  3.33    4.56    6.77     8.5       8.5      8.5     8.5     8.5

---

> > > ### Comment · AnonReviewer3 · 2018-11-02
> > > **Please specify the exact differences**
> > >
> > > I appreciate the additional experiments, thanks! Could you specify exactly what the difference between NES and your attack is? If I understand you correctly, then the difference is (1) you are using a smaller standard deviation for sampling and (2) you don't perform clipping. However, the standard deviation is merely a hyperparameter of NES and should be tuned for optimal attack efficiency. Second, the clipping is necessary in any real world scenario where you don't have full access to the model but can only query it with images, right?

---

> > > > ### Author Response · Authors · 2018-11-02
> > > > **Clarification**
> > > >
> > > > Thanks for asking. We will clarify our previous responses (mainly the paragraph below) by answering three of your questions.
> > > >
> > > > ----------------------------------------
> > > > On the algorithmic aspect, both ours and Ilyas et al. (2018)’s employ NES as the optimization algorithm. However, we arrive at it via different routes and for different purposes. We assume a probabilistic generation process of the adversarial examples (Steps 1–4, Section 2), which finds an adversarial example by a one-step addition to the input. In contrast, Ilyas et al. (2018)’s modeling assumption is that an adversarial example can be found by PGD, which iteratively updates the original input with a small learning rate until it becomes adversarial. To this end, we use NES to estimate the parameters of the distribution, while Ilyas et al. (2018) use NES to replace the true (stochastic) gradients in PGD. We contend that, due to the non-differentiable clip and projection operations and the fairly large Gaussian covariance, NES is *not* an efficient (and possibly a biased) estimator of the true gradients — we are running experiments to empirically verify if this is true or not.
> > > > --------------------------------------------------------------
> > > >
> > > > == Q1: specify exactly what the difference between NES and your attack is? ==
> > > >
> > > > Using our notation, the pseudo code below sketches our algorithm and Ilyas et al. (2018)’s.
> > > >
> > > > Ours, which searches for the Gaussian from which more than one adversarial examples can be generated.
> > > > Iterate until convergence:
> > > > 1. Draw a sample {\epsilon} from the normal distribution
> > > > 2. Transform it to a sample of Gaussian by {z=\theta + \sigma * \epsilon}
> > > > 3. Generate current adversarial examples from {z} by steps 1–4
> > > > 4. Compute the losses {J(z)}
> > > > 5. Compute the search gradients {g} by equation (5)
> > > > 6. Update the Gaussian mean: \theta = \theta - r * g
> > > > Return \theta
> > > >
> > > > Ilyas et al. (2018)’s, which searches for a single adversarial example.
> > > > Iterate until convergence:
> > > > 1. Draw a sample {e} from the normal distribution
> > > > 2. Transform it to a sample of zero-mean Gaussian by {z=0 + \sigma * \epsilon}
> > > > 3. Generate current adversarial examples by {x + z} and {x - z}
> > > > 4. Compute the losses {J(z)}
> > > > 5. Compute the search gradients {g} by equation (5)
> > > > 6. x = Projection(x - r * sign(g))
> > > > Return x
> > > >
> > > >
> > > > The differences start from the second line, where we transform the normal sample to a sample of the Gaussian N(\theta, sigma^2) while Ilyas et al. (2018) transform it following a zero-mean Gaussian N(\theta, sigma^2).
> > > >
> > > > Line 3: The difference is on how to generate the adversarial examples.
> > > >
> > > > Line 4: Slightly different loss functions are used in the two methods. This is not vital.
> > > >
> > > > Line 5 is the same for the two methods.
> > > >
> > > > Line 6: While we update the Gaussian mean by a gradient descent step, Ilyas et al. (2018) update the adversarial example by PGD.
> > > >
> > > > == Q2: a smaller standard deviation for sampling ==
> > > > By using the same setting for the NES component of our algorithm and Ilyas et al. (2018)’s, including the same sample size and standard deviation, we obtain the comparison results below. Ours still performs better. We will complete the experiments with all the defense methods studied in our paper.
> > > >
> > > > Table 1: Success rate on attacking Randomization (ImageNet)
> > > > # of iterations          30       90       150      210       270    300     360     400
> > > > ours                         21.54  78.58  90.02   95.41   95.5    95.5    95.5    95.5
> > > > Ilyas et al. (2018)'s  20.5    46.37  53.33   53.33   53.33  53.33  53.33  53.33
> > > >
> > > >
> > > > == Q3: you don't perform clipping ==
> > > > We did perform clipping in steps 1--4 of the paper, where we generate adversarial examples from a Gaussian distribution. In contrast, Ilyas et al. (2018)’s performs the clipping of gradients due its employment of PGD attack.

---

### Public Comment · (anonymous) · 2018-11-02
**Questions**

Hi, it looks very interesting.

However, I have a few questions.

(1) Could you specify the threat model? For example, I could not find what substitute models are used to generate adversarial examples. What black-box setting did you use?

(2) I think you don't actually evaluate your attack on Madry et al. (2018). THERM-ADV did not technically use PGD adversarial examples described in Madry et al. (2018), but use LS-PGA examples described in Buckman et al. (2018). In addition, Athalye et al. (2018) argued that THERM-ADV is significantly weaker than Madry et al. (2018) since it is trained against the LS-PGA attacks.  Therefore, The argument in your paper, "Athalye et al. (2018) find that the adversarial robust training (Madry et al., 2018) can significantly improve the defense strength of THERM." may be wrong.

---

> ### Public Comment · ~Nicholas_Carlini1 · 2018-11-02
> **Regarding thermometer encoding**
>
> The statement as it is written is technically correct. Thermometer encoding by itself is no more robust than a standard neural network. Adding adversarial training to thermometer encoding confers some amount of robustness, but less than standard adversarial training.
>
> So whether or not adversarial training can "significantly improve the defense strength of THERM" depends I guess on your definition of "significantly". In Athalye et al. (2018) we find this difference to be ~20% at eps=8 and ~40% at eps=4.

---

> > ### Public Comment · (anonymous) · 2018-11-02
> > **Clarification**
> >
> > Thanks for your comment!
> >
> > I clarify my concern more clear. I was trying to say that THERM-ADV of Buckman et al. (2018)  should not be cited as Madry et al. (2018) for evaluation since standard adversarial training, Madry et al. (2018), is more robust than THERM-ADV.

---

> > > ### Author Response · Authors · 2018-11-02
> > > **Answers to (1) & (2)**
> > >
> > > Regarding (2), thank @Nicholas for the catch! We will cite both (Buckman et al., 2018) and (Madry et al., 2018) in the revised paper. It is worth noting that vanilla PGD does not apply to Therm. As a result, the LS-PGA enhanced Therm-Adv is probably one of the best one can do in order to apply Madry et al. (2018)'s defense principle to Therm.
> > >
> > > Regarding (1), you may consider the Gaussian distribution along with the steps 1--4 in the paper as the threat model. We do not use any substitute network in our approach. The black-box setting: We query a black-box network by an input and obtain its output probability vector.  This setting is as standard as many existing works'.

---

> > > > ### Public Comment · (anonymous) · 2018-11-02
> > > > **Vanilla PGD training**
> > > >
> > > > Thanks for the answer. I still have a question. Vanilla PGD training is more robust than Therm-Adv.
> > > > For (2), why do you not use vanilla PGD training instead of Therm-Adv?

---

> > > > > ### Author Response · Authors · 2018-11-02
> > > > > **Response to "Vanilla PGD training"**
> > > > >
> > > > > Since Therm discretizes the input, it prevents one from simply applying the gradient projection in PGD, not mentioning that the gradients are estimated through other methods like BPDA or DGA (Buckman et al., 2018). Did you mean that DGA is a more faithful application of (Madry et al., 2018)'s defense to Therm than LS-PGA? However, DGA has been shown a weak attack so it likely cannot lead to strong defense (e.g., called Therm-Adv-DGA) either.

---

> > > > > > ### Public Comment · (anonymous) · 2018-11-02
> > > > > > **standard adversarial training**
> > > > > >
> > > > > > You don't have to use the Thermometer encoding anymore. I mean the pure vanilla PGD training which Athalye et al. (2018) were unable to defeat.

---

> > > > > > > ### Author Response · Authors · 2018-11-02
> > > > > > > **Will do**
> > > > > > >
> > > > > > > Oops, I misunderstood your earlier question.. We are running experiments against the vanilla PGD defended CNN. As Athalye et al. (2018) did not release this very strong model, we had to train it ourselves. Actually, we will ask them for that model by email now.. Stay tuned please.

---

> > > > > > > > ### Public Comment · (anonymous) · 2018-11-02
> > > > > > > > **Public code**
> > > > > > > >
> > > > > > > > FYI, Madry et al. released publicly available pre-trained weights.
> > > > > > > >
> > > > > > > > https://github.com/MadryLab/mnist_challenge
> > > > > > > > https://github.com/MadryLab/cifar10_challenge

---

> > > > > > > > > ### Author Response · Authors · 2018-11-02
> > > > > > > > > **Got it**
> > > > > > > > >
> > > > > > > > > Got it. Thanks for the pointers!

---

> > > > > > > > > > ### Author Response · Authors · 2018-11-23
> > > > > > > > > > **Results**
> > > > > > > > > >
> > > > > > > > > > Here are the success rates on attacking the vanilla PGD training (Madry et al., 2018) on CIFAR10:
> > > > > > > > > > BPDA: 46.9%
> > > > > > > > > > Ours: 47.9%
> > > > > > > > > >
> > > > > > > > > > The classification accuracy of the PGD-defended network is 87.3% on CIFAR10.
> > > > > > > > > >
> > > > > > > > > > Conclusion 1: The vanilla PGD training is strong.
> > > > > > > > > > Conclusion 2: The vanilla PGD training sacrifices the performance to certain degree on the original classification task, so do some other defense techniques (cf. Table 1 in the PDF).

---

### Author Response · Authors · 2018-11-23
**Summary of changes in the new PDF**

Denote by QL (Ilyas et al., 2018)’s approach. We have added to the revised PDF
+ new results on attacking Adv-Train (Madry et al., 2018) (Table 1),
+ a new paragraph to draw readers’ attention to QL upfront (cf. the highlighted text in the introduction),
+ new results of QL on attacking the 10 defense methods (Table 1),
+ a new section (Section 3.1.3) to carefully investigate the factors that contribute to the inferior performance of QL algorithm. The results reveal that, in order to improve its attack success rates, it is vital to get rid of PGD (projection and the sign of the gradients), which is the foundation upon which QL is built, and meanwhile to couple the $\ell_infty$ clip with the tanh transformation.

Thanks to the careful experimental investigation, we make the following conclusion.

1) QL is hinged on the white-box PGD attack --- in terms of methodology, it is actually closer to [1] than ours because both QL and [1] essentially approximate the gradients for PGD. As a result, the quality of the estimated gradients in QL is a big deal. Unfortunately, ES does not give rise to stable gradients due to the sampling step and PGD’s projection and sign functions, especially when the gradients are "obfuscated". On the contrary, we do not employ any white-box attack methods at all in developing our algorithm. The Gaussian mean is more important than the gradients in our approach. Whereas ES is a natural choice to search for the Gaussian mean, some derivative-free methods [2] are also good alternatives.

2) The seemingly subtle algorithmic distinction between QL and ours actually leads to significantly different attack success rates. In order to improve QL’s performance, it is vital to remove PGD, the foundation upon which QL is built.

[1] Anish Athalye, Nicholas Carlini, and David Wagner. Obfuscated gradients give a false sense of security: Circumventing defenses to adversarial examples. arXiv preprint arXiv:1802.00420, 2018.
[2] Luis Miguel Rios and Nikolaos V Sahinidis. Derivative-free optimization:  a review of algorithms and comparison of software implementations. Journal of Global Optimization, 56(3):1247–1293,2013.

---

### Meta-Review · Area_Chair1 · 2018-12-13
**the only favorable review does not make a convincing argument to accept the paper**

**Confidence:** 5
**Recommendation:** Reject

**Metareview:**

Although one review is favorable, it does not make a strong enough case for accepting this paper. Thus there is not sufficient support in the reviews to accept this paper.

I am recommending rejecting this submission for multiple reasons.

Given that this is a "black box" attack formalized as an optimization problem, the method must be compared to other approaches in the large field of derivative-free optimization. There are many techniques including: Bayesian optimization, (other) evolutionary algorithms, simulated annealing, Nelder-Mead, coordinate descent, etc. Since the method of the paper does not use anything about the structure of the problem it can be applied to other derivative-free optimization problems that had the same search constraint. However, the paper does not provide evidence that it has advanced the state of the art in derivative-free optimization.

The method the paper describes does not need a new name and is an obvious variation of existing evolutionary algorithms. Someone facing the same problem could easily reinvent the exact method of the paper without reading it and this limits the value of the contribution.

Finally, this paper amounts to breaking already broken defenses, which is not an activity of high value to the community at this stage and also limits the contribution of this work.